# Wi-Alarm: Low-Cost Passive Intrusion Detection Using WiFi

**DOI:** 10.3390/s19102335

**Published:** 2019-05-21

**Authors:** Tao Wang, Dandan Yang, Shunqing Zhang, Yating Wu, Shugong Xu

**Affiliations:** Shanghai Institute for Advanced Communication and Data Science, Key laboratory of Specialty Fiber Optics and Optical Access Networks, Joint International Research Laboratory of Specialty Fiber Optics and Advanced Communication, Shanghai University, Shanghai 200444, China; twang@shu.edu.cn (T.W.); yang03@shu.edu.cn (D.Y.); shunqing@shu.edu.cn (S.Z.); shugong@shu.edu.cn (S.X.)

**Keywords:** WiFi, device-free passive detection, CSI, low cost

## Abstract

In this paper, we present a WiFi-based intrusion detection system called Wi-Alarm. Motivated by our observations and analysis that raw channel state information (CSI) of WiFi is sensitive enough to monitor human motion, Wi-Alarm omits data preprocessing. The mean and variance of the amplitudes of raw CSI data are used for feature extraction. Then, a support vector machine (SVM) algorithm is applied to determine detection results. We prototype Wi-Alarm on commercial WiFi devices and evaluate it in a typical indoor scenario. Results show that Wi-Alarm reduces much computational expense without losing accuracy and robustness. Moreover, different influence factors are also discussed in this paper.

## 1. Introduction

Intrusion detection is a dynamic process of monitoring whether there exists any entity breaking into a given area, then making alarms if necessary. It gains more and more attention and has great potential in many applications, such as border protection [1], smart home [2], elderly health care [3]. Several approaches to intrusion detection have been proposed, among them, WiFi-based intrusion detection has superiority for its extensive coverage which means low deployment cost. In general, WiFi-based systems can be divided into device-based active approaches and device-free passive (DFP) [4] approaches. The former requires a target to carry a mobile device and turn on its WiFi, then the target is identified by that device. Obviously it is not suitable for intentional intruders. Therefore, the DFP approach is focused in this paper.

Most WiFi-based DFP approaches take advantage of the transmission characteristics of opportunistic WiFi signal in wireless channels, such as a received signal strength indicator (RSSI) [5] and channel state information (CSI) [6,7,8,9], to capture changes in a region [10]. RSSI suffers from dramatic performance degradation due to multipath fading [11]. Compared with RSSI, CSI not only provides more stable and reliable amplitude information than RSSI, but also reveals phase information.

Among existing CSI-based systems of intrusion detection, considerable effort has been devoted to data preprocessing. Some do preprocessing to gain useful phase information [8,9], others for de-noising [6,7]. However, through our analysis and experiment observations, we think that although the raw phase of CSI is meaningless [12], the raw amplitude of CSI is informative and relatively robust.

In this paper, we propose a novel and real-time system to detect intrusion. We call it Wi-Alarm because WiFi signals work as an alarm to warn people of intrusion here. Firstly, CSI samples are collected with CSI Tool [13]. Then, we omit data preprocessing and then extract features from amplitude of raw CSI. Next, an algorithm based on the support vector machine (SVM) is introduced to give detection results (moving human presence or absence). Finally, we consider different influence factors to improve system performance including size of sliding window, multiple-input multiple-output (MIMO), and line-of-sight (LOS)/not line-of-sight (NLOS), which means LOS can be blocked by human motion or not.

Our main contributions are summarized as follows:We focus on the real-time performance of the whole system, which is neglected in previous research. In order to save computational expense significantly, we omit data preprocessing and extract features from raw CSI directly for intrusion detection, based on our analysis of the data preprocessing part in other systems and experimental verification. Given that real-time performance is critical, any effort to reduce computational expense is worthwhile if high detection accuracy can be maintained.We propose a simple but robust feature couple for intrusion detection where low latency is necessary. The feature couple could be promising for commercial use.Dynamic walking speeds are considered in our training phase. This improves Wi-Alarm’s sensitivity to human motion and enhances detection accuracy even when an intruder walks quite slowly.Different influence factors including MIMO, LOS/NLOS, size of sliding window and walking speed are discussed to guide better implementation of our system to improve the detection accuracy and robustness.

The rest of this paper is organized as follows. Section 2 introduces related work in DFP detection using WiFi. In Section 3, the overview of Wi-Alarm is presented. This is followed by the methodology of our system and the detailed design in Section 4. Then in Section 5, we describe the implement and experiment results of Wi-Alarm, different influence factors are discussed in this section. Finally, Section 6 concludes this paper and suggestions are made for future work.

## 2. Related Work

In this section, we review the most related works on WiFi-based DFP detection.

### 2.1. RSSI-Based

Kosba et al. proposed a system named RASID [5], in RASID, the sample variance of RSS is used as the selected feature, then non-parametric statistical anomaly and profile update techniques are applied to capture changes in the environment. RASID is robust, but the sampling rate of 1 sample/s is too low, resulting in too long a time period for data collection. Yoshida et al. collected RSSI values at existing WiFi devices inserted with clock-synchronized Raspberry Pi device [14], with the outliers removed, pre-processed RSSIs are applied to two regression-based approaches for estimating the number of people in the scenario. The non-real-time experimental results showed that support vector regression-based method is better for estimating the presence/absence of people than linear regression-based method. Depatla et al. proposed a crowd counting system using only one pair of personal computer (PC) and access point (AP) with 802.11g WiFi cards, the system is based on the mathematical expression they deduced where the impact of human movement on RSSI values is divided into two parts: blocking LOS and scattering effects [15]. Kullback-Leibler divergence between the theoretical and experimental probability mass function (PMF) of RSSI is applied to estimate the total number of people.

Generally, RSSI-based systems need to deploy more WiFi sensing nodes to reduce the influence of multipath effect for higher accuracy.

### 2.2. CSI-Based

CSI-based systems can be divided into two categories: using the amplitude of CSI and using both amplitude and phase of CSI. In the first class, robust passive motion detection (R-PMD) [6] passes the CSI sequence through a data preprocessing module, then extracts the variance distribution of the CSI sequence and utilizes the earth mover’s distance (EMD) to infer the degree of abnormality. Speed independent device-free entity detection (SIED) [7] extracts the distribution of the variance of variances of CSI among all the subcarriers as feature and leverages a probability technique hidden Markov model (HMM) as the classifier to make it more accurate in human detection, it performs well when the moving speed is very slow. WiGarde [16] was proposed to detect an intruder through door or window for home safety, in WiGarde, a naive Bayesian classifier was used to eliminate bad stream caused by surrounding electromagnetic noise, wavelet was used to get the width of the dynamic time window for extracting the best feature, one-class SVM was adopted to classify human intrusion. In [17], Zhou et al. applied support vector classify (SVC) to solve the presence detection and applied support vector regression (SVR) to solve the localization problem through regression. They applied density-based spatial clustering of applications with noise (DBSCAN) on CSI data for de-noising and principal component analysis (PCA) for feature extraction. Experiments show that, compared with Bayesian algorithm, SVC performed 16.6% better in the meeting room.

In the second class, Kun et al. firstly employed a liner transformation to eliminate the significant random noise in CSI phase in passive detection of moving humans with dynamic speed (PADS) [8], it uses the maximum eigenvalues of covariance matrixes from normalized amplitude and phase as features, then SVM is introduced for different states estimation, PADS is able to accurately detect human movements of dynamic speed. Ref. [9] uses the same transformation as in [8] to eliminate the shift of phases of different subcarriers, it introduces the effect size to measure the change of phase as a feature and it defines two reference points for the short-term case (SES) and the long-term case (LES) to detect if someone is walking in indoor room and if someone is walking continuously respectively. However, the walking area is limited in this system, so technically, detection targets are not free. Ding et al. derived robust phase difference from the processed phase in neighbouring antennas after raw CSI preprocessing [18], extracted the covariance matrixs of normalized CSI amplitude and phase difference as features, three machine learning classification algorithms are applied for motion detection and the best results were achieved with SVM.

The fine-grained information, CSI, not only makes it possible to sense human body, but also could recognize more subtle activity. WiFall [3] proposed by Han et al. for fall detection compares one-class support vector machine method with random forest algorithm and it is proved by experiment results that the latter one works better. Wang et al. proposed CSI-based human activity recognition and monitoring system (CARM) [19] to recognize multiple human activities based on two theoretical models, CSI-speed model and CSI-frequency where the relation between path length and CSI power is quantified and the recognition accuracy is 96%. Zhang et al. [20] introduced the Fresnel zone model in optical into the transmission of radio waves to realize millimeter-scale detection of human respiration and discussed the theoretical sensing limit of WiFi signal.

Despite that many works have investigated CSI for device-free detection, the computational expense of the whole system is lack of study, and usually they just consider few influence factors. In this paper, we propose a low cost system for intrusion detection without losing accuracy. Moreover, different factors including MIMO, LOS/NLOS, size of sliding window and dynamic walking speeds are considered in this paper.

## 3. Overview

In this section, we briefly introduce the architecture and flowchart of our system.

### 3.1. System Architecture

As shown in Figure 1, former works generally contain following modules: CSI collection, data preprocessing, feature extraction, comparing real-time test data with static fingerprint, and finally getting detection results. As marked, our biggest innovation is omitting the data preprocessing module which usually includes outlier removal, interpolation, sanitization, and de-noising. Therefore, our system mainly contains three parts: data collection, feature extraction, and intrusion detection, which includes SVM classifier generation and anomaly detection.

### 3.2. System Flowchart

Our system works as Figure 2 shows, compared with Figure 1, it is obvious that data preprocessing is omitted in our system, so Wi-Alarm has lower overhead for omitting the operation of outlier removal, interpolation and so on. Our system works according to the following process. Firstly, during the CSI collection phase, raw CSI samples are measured and collected by commercial WiFi devices. Then, our system contains two phases: a training phase and a monitoring phase. During the training phase, data collection contains four cases: static environment with nobody and one person moving at three different speeds (0.2 m/s, 0.7 m/s and 1.5 m/s) casually and freely. Since we omit data preprocessing, features are extracted from raw CSI amplitude directly. Based on these, a SVM classifier is generated and two patterns are constructed including a static pattern and a dynamic pattern. This enhances our system’s robustness to intruder’s moving speed. During the monitoring phase, features extracted from unknown CSI samples are applied to the well-trained SVM classifier to get detection results. According to the result, Wi-Alarm decides to alarm or not.

## 4. Methodology

In this section, we give the details of Wi-Alarm system. Generally, we aim to propose a low-latency intrusion detection system with high accuracy. Therefore, the main indicators of the whole system are the performances of real-time and accuracy. In order to improve the real-time performance, we devote much effort to reduce the computational cost of each step without reducing detection accuracy.

### 4.1. Data Collection

CSI is a channel attribution of communication link which describes attenuator factor on each transmission path including scattering, environmental weakness, distance attenuation, etc. Different from RSSI, which demonstrates superimposition of signals, CSI could be considered as the channel response depicting the amplitudes and phases of 30 subcarriers, so CSI is finer-grained and more stable.

As shown in Figure 3, our CSI collection system required a laptop configured with Intel WiFi 5300, which served as a detection point (DP). The DP sent internet control message protocol (ICMP) echo requested packages to the WiFi router which works as an AP and receives ICMP echo reply packages if the connection succeeded [13]. This process was modeled as:(1)Y=HX+N,
where *X* and *Y* are the transmit and receive vector respectively while *N* is the ambient noise vector, then *H* is the CSI required. The estimated *H* could be expressed as:(2)$$\hat{H}=\frac{Y}{X}.$$

For each sample, the CSI can be expressed as: (3)$$H={\left[H\left(f_{1}\right),\ldots ,H\left(f_{i}\right),\ldots ,H\left(f_{N_{sub}}\right)\right]}^{T},i=1,2,\ldots ,30.$$
where $N_{sub}$ means the number of subcarriers (equals to 30) and $f_{i}$ is the frequency of *i*-th subcarrier. In 802.11n protocol, it can range from 2.4 GHz to 2.4835 GHz. For each OFDM subcarrier, CSI is defined as:(4)$$H\left(f_{i}\right)=\left|H\left(f_{i}\right)\right|e^{j∠H(f_{i})},$$
where $\left|H(f_{i})\right|$ means the amplitude and $∠H(f_{i})$ means the phase of CSI.

Since the CSI collection system is a MIMO system, suppose there are *M* transmit antennas and *N* receive antennas (NIC 5300 had three antennas), then there were $N_{ch}=M\times N$ WiFi channel links. In each link, 30 of 56 subcarriers were sampled and the number of packages increased over time. The CSI sequence in *n*-th link could be expressed as a $30\times N_{pkt}$ complex matrix:(5)$$|H^{n}|=\left[|H_{1}^{n}\left|,\ldots ,\right|H_{i}^{n}\left|,\ldots ,\right|H_{N_{pkt}}^{n}|\right],i=1,2,\ldots ,N_{pkt},$$
where $N_{pkt}$ is number of CSI package and $|H_{i}^{n}|$ is CSI amplitude of *i*-th package in *n*-th link.

Usually, we will not receive CSI packages endlessly, given a specific sliding window with length *W*, then $N_{pkt}$ equals *W*. Moreover, the time of data collection depended on the length of *W*. In this part, the length of sliding window was significant. If the size was too small, the CSI samples were too few to calculate an effective feature, which will reduce the detection accuracy. Otherwise, larger size means more time to collect more CSI samples, that will undermine the real-time performance of the system. In our system, through experiment analysis, it was appropriate to set to transmit 100 packages per second, i.e., CSI sampling frequency is 100 Hz. Therefore, the time needed for single CSI collection can be denoted as:(6)$$t=\frac{W}{100}.$$

### 4.2. Feasibility of Preprocessing Omission

In Wi-Alarm, we extracted features from raw CSI data without preprocessing in order to save computational cost for better real-time performance. There was no doubt that omitting preprocessing will reduce much computational cost, in this part, we discuss the feasibility of preprocessing omission.

In previous works, preprocessing was important to minimize environmental noise and usually done between raw CSI collection and feature extraction. This motivated us to think: how to understand the dynamic fluctuation, which is considered as noise before? With human motion, CSI values fluctuate during transmission. More dynamic fluctuation would make our detection easier. However, preprocessing in previous systems would suppress signal fluctuation which usually carries an intruder’s motion information.

In order to validate the feasibility of preprocessing omission, we repeated the data preprocessing in R-PMD [6], a well-known robust DFP motion detection system. Firstly, Hampel identifier was utilized to remove outliers. In one sequence, samples falling out of [μ−3∗δ,μ+3∗δ] were identified as outliers where μ is median, δ is median absolute deviation (MAD). When $\left|H\right|=\left[|H_{1}\left|,\ldots ,\right|H_{i}\left|,\ldots ,\right|H_{W}|\right]$, the μ and δ are expressed as follows: (7)μ=Median(|H|),
(8)$$\delta =\frac{1}{W}\sum_{i=1}^{W}|\left|H_{i}\right|-\mu |.$$

Secondly, 1D linear interpolation algorithm is applied to fill the spaced samples. Lastly, environment noise was reduced by weighted moving average (WMA), CSI of the *m*-th package was averaged by previous (m−1) CSI packages, i.e.,
(9)$$|H_{m}^{{}^{\prime }}|=\frac{1}{m+(m-1)+\ldots +1}*\left[m*|H_{m}\left|+\left(m-1\right)*\right|H_{m-1}\left|+\ldots +1*\right|H_{1}|\right]$$

Then subsequent features were extracted from $|H_{m}^{{}^{\prime }}|$ in R-PMD.

CSI values with or without preprocessing behaved quite differently. Figure 4 and Figure 5 show that raw CSI values fluctuate significantly with human motion, while preprocessing smooths these original fluctuations.

In conclusion, preprocessing omission is feasible. We decided to omit this step and extract features from raw CSI.

### 4.3. Feature Extraction

Feature extraction plays an important role in intrusion detection. The features we choose should meet the following requirements.

The features should be sensitive to human presence. The features extracted from the CSI of static environment should be quite different from the ones extracted from the CSI with human motion.The features should be easy to calculate. For real-time consideration, less computation cost on feature extraction would be better.

In our system, only CSI amplitude was applied. Compared with raw CSI amplitude, raw CSI phase behaved with much randomness due to environment noise and an unsynchronized time clock between transmitter and receiver. Figure 6 shows that amplitude outperforms phase in stability, so CSI phase was deprecated here.

We propose a simple but robust feature couple for Wi-Alarm in consideration of low latency. Various features have been applied in former research for detection, such as mean, variance, and distribution distance. However, the computational expense of distribution distance is much higher, which makes it inappropriate for real-time processing. Motivated by our experiment observations that, compared with static environment, human motion leads to higher CSI amplitudes with greater variances, we proposed the mean and variance of CSI amplitude to be a couple of good indicators for intrusion detection. Denote the feature couple as F[α,β], where α represents the mean and β represents the variance of CSI amplitude respectively.

In a fixed WiFi link, one feature couple *F* is derived from one CSI sequence, given a specific sliding window *W*. Specifically, we calculated the mean/variance of all packages for one subcarrier firstly, then got the mean of all 30 subcarriers as the final α and β, i.e.,
(10)$$\alpha =\frac{1}{N_{sub}}\sum {}_{i=1}^{N_{sub}}\left[\frac{1}{W}\sum {}_{j=1}^{W}\left|H_{j}\right|\right],$$
(11)$$\beta =\frac{1}{N_{sub}}\sum {}_{i=1}^{N_{sub}}\left[\frac{1}{W}\sum {}_{m=1}^{W}{\left(|H_{m}|-\frac{1}{W}\sum {}_{n=1}^{W}\left|H_{n}\right|\right)}^{2}\right].$$

Then one feature couple *F* of one CSI sequence was calculated, it could be exploited in later SVM classifier generation and real-time intrusion detection.

### 4.4. Classifier

With features we extracted, we needed to conduct an appropriate classifier on preliminary measurements collected from several cases. There were mainly two categories of methods that can be adopted for calibration-free detection: threshold-based and cluster-based. The former distinguish different states based on the threshold value which is gained from pre-collected data. The latter classifies different clusters as different states by comparing the centre distance of each cluster. Although environment calibration and threshold training were avoided in the cluster-based method, it assumed that in each group of measurements, there were at least two states involved. Otherwise, it would lead to one cluster or several clusters corresponding to a same state, which means miss or false detection. Consequently, the threshold based scheme was adopted in Wi-Alarm.

Support vector machine (SVM), as a low cost threshold based classification, is one of the most popular machine learning techniques. A considerable number of CSI-based detection systems [8,14,16,17,18] have applied SVM to classify different states and achieved high accuracy. Experiments show that SVM outperforms other machine leaning method like k-nearest neighbor (KNN) [17] and Bayesian algorithm [18]. Therefore, SVM algorithm was adopted for satisfying the system requirements for real-time and high detection accuracy.

In this paper, we adopt the radial basis function (RBF) kernel (a.k.a. Gaussian kernel) as the kernel function so that a non-linear regression model can be constructed. This SVM has two such training factors: *C* which controls overfitting of the model, and γ which controls the degree of nonlinearity of the model. In our system, we set C=0.8 and γ=20.

The training phase in our system was different from former works. Previous works only contain static fingerprint construction. In Wi-Alarm, shown in Figure 2, data of four different cases was applied to train our SVM classifier which contains a static pattern and a dynamic pattern. This helps reduce the adverse effect from dynamic walking speed on detection accuracy. Then in the monitoring phase, the feature couple extracted from real-time data was applied to the trained SVM classifier. Therefore, a detection result was obtained. If an intrusion was detected, Wi-Alarm would raise an alarm.

## 5. System Evaluation

In this section, the experiment setup and evaluation of Wi-Alarm are illustrated.

### 5.1. Experimental Setup

To evaluate the performance of Wi-Alarm, we conducted experiments on commercial devices. The configuration of our system is shown in Figure 7 and Table 1. It should be noticed that the antennas of NIC 5300 were isometrically arranged for stable receiving performance. The experiments are conducted in an office (2.5 m × 4 m) as shown in Figure 8.

As we mentioned that the train phase contained four cases, for each case, we conducted three measurements and each lasted for 2 min. In dynamic cases, the person can walk freely and easily without rules or restrictions. The detection results would be divided into two categories: with/without intrusion as shown in Figure 9.

Since the influence of LOS/NLOS conditions were considered, the AP and DP were placed at two heights: 0.8 m and 2 m. In the condition of 0.8 m, the movement of a human would block LOS while not in the condition of 2 m.

### 5.2. Performance Results

We use following four metrics to evaluate the performance of proposed Wi-Alarm system:True negative (TN) rate: the probability that the human motion is correctly detected.True positive (TP) rate: the probability that the static environment is correctly classified.False negative (FN) rate: the probability that the human motion is incorrectly detected.False positive (FP) rate: the probability that the static environment is incorrectly classified.

We calculated TN/TP/FN/FP rates of all conditions and the discussion of our experiment results are shown as follows:

Impact of preprocessing omission: among all the experiments with LOS and the size of sliding window W=100, the TN and TP rates are both higher than 99% no matter we use one antenna or combination of different antennas. R-PMD [6] was repeated in our experiments. The detection results of these two systems are shown in Figure 10. Both of these two systems achieved high accuracy, the TP and TN rates of Wi-Alarm were higher, which are up to 100% and 99.06% respectively. In addition, the receiver operating characteristic (ROC) curve of our classifier is shown in Figure 11, the result proves that the classifier we choose is trained well and achieves excellent performance. The runtimes of these two systems differ a lot in Table 2 (taking 10,000 CSI samples as an example, running in Matlab). Because of the omission of data pre-processing, the runtime was greatly shortened. During the classification, R-PMD needed more computational expense for EMD metric calculation.

Impact of sliding window size: generally, choosing the size of the sliding window is critical. If the size is too small, the CSI samples are too few to calculate an effective feature. Otherwise, larger size means more time to collect more CSI samples, undermining the real-time performance of the system. According to our experiments, in Figure 12, it can be verified that TP rate is proportional to the size of sliding window while TN rate is not. Also, FP rates were generally higher that FN rates and it was minimized when the window size was set to 100. In summary, the most appropriate window size of our system was 100. Therefore, in the following comparative experiments, the window size was fixed to 100.

Impact of different antennas: we considered four cases here: each individual antenna and the combination of all three antennas. The way of combination is taking the mean of all three antennas. Figure 13 shows that the performances of each case are similar, single antenna even achieves lower FN and FP rates. This means using multiple antennas was unnecessary, because the combination of MIMO needed more computational expense.

Impact of LOS/NLOS: we changed the height of AP and DP to create two different detection environments: LOS and NLOS, the former one means that LOS would be blocked by human motion during the monitoring phase by low placement of AP and DP, while the latter one would not by relatively high placement. Figure 14 shows that NLOS leads to lower TN rate and much higher FP rate obviously. Therefore, LOS setting was preferred in our system and we should make sure the height of AP and DP was not too low or too high. The height of 0.8 m would be a suitable parameter setting without being affected by furniture or missing intrusion.

Impact of SVM parameters: the selection of SVM parameters also had a significant impact on the detection rate of Wi-Alarm. Through our experiment evaluations, on one hand, higher detection rates were achieved with bigger *C*, as Figure 15 shown. However, *C* controls overfitting of the model, so it should not be too big and 0.8 was a good setting. On the other hand, as Figure 16 shows, the TN rate decreases with the increase of γ. Meanwhile, γ controls the degree of nonlinearity of the model and bigger γ was preferred in this respect. Therefore, 20 was a good setting for γ. In conclusion, we set C=0.8 and γ=20 for the best performance of Wi-Alarm.

## 6. Conlusions

In this paper, we present a passive intrusion detection system called Wi-Alarm, which is based on CSI using WiFi. In our system, attacks are performed when an intruder is entering the monitoring area unexpectedly wherever he comes from. Wi-Alarm is proposed mainly to distinguish two states (human presence and absence) and detect whether there exist any people in the area of interest. It could be applied in the field of safety monitoring.

Motivated by our observation that raw CSI of WiFi is sensitive enough to detect human motion, we omit data preprocessing and utilize raw CSI amplitude for feature extraction, SVM algorithm is used as a classifier in our system. We have conducted extensive experiments, evaluate results to validate the robustness, high detection rate and real-time of Wi-Alarm. Besides, different influence factors are considered in our experiments which will guide better implemention of Wi-Alarm.

However, there still exists some limitations. Firstly, when installed in a different scenario, our system needs to be trained again to find a suitable threshold. Moreover, usually there already exists someone in the given area which means the original environment is not static before any intrusion occurs. Consequently, we should explore new features to model the motion introduced by the intruder coming from the outside aiming to distinguish it from the motion inside.

Future work will focus on exploring novel and effective features to enhance the adaptability of our system and the functions of detecting more types of attacks. 

## Figures and Tables

**Figure 1 sensors-19-02335-f001:**
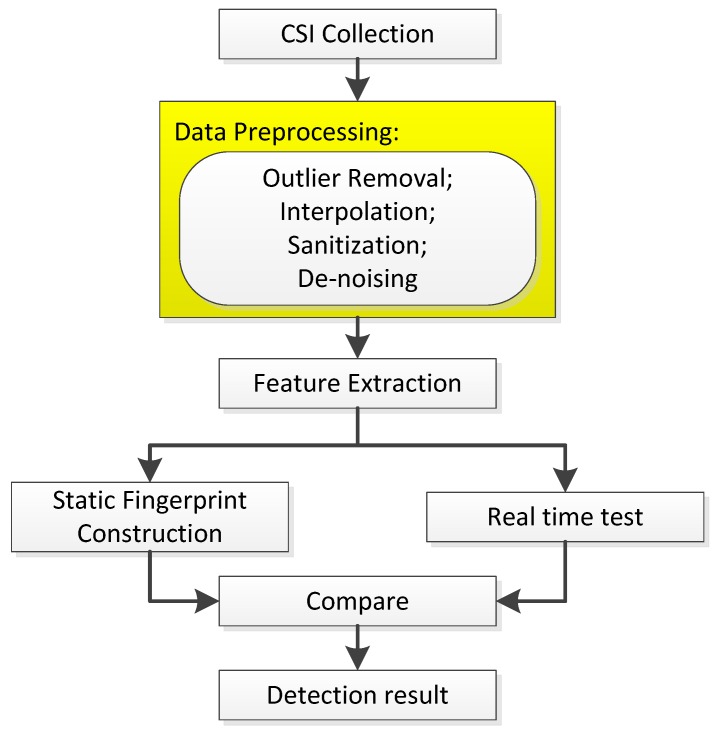
Common architecture of former works.

**Figure 2 sensors-19-02335-f002:**
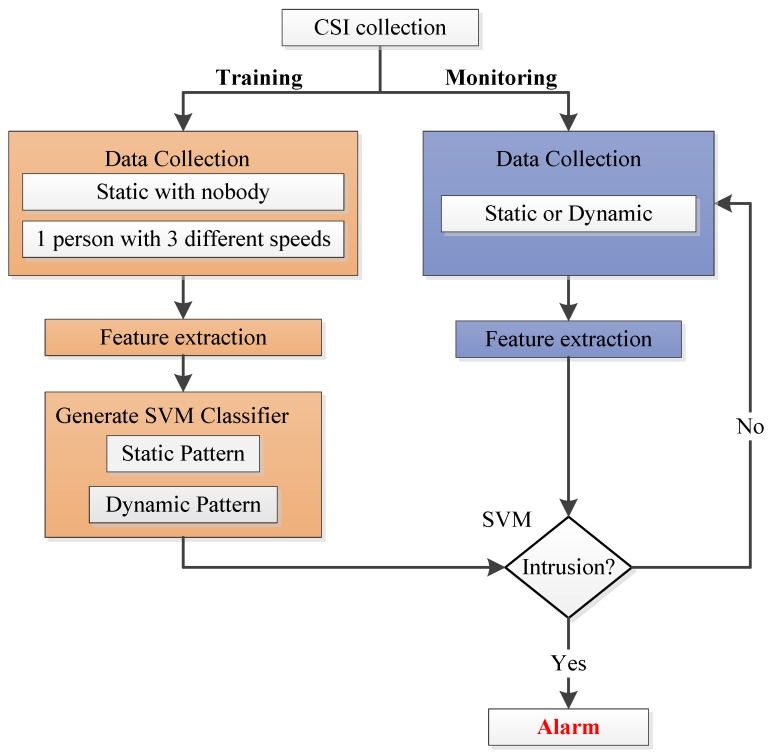
System flowchart.

**Figure 3 sensors-19-02335-f003:**
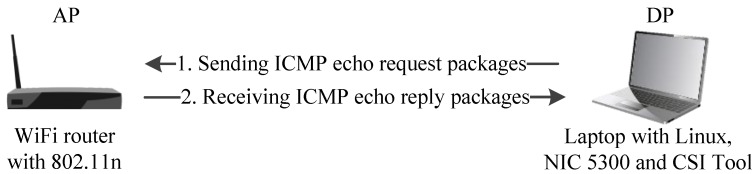
Channel state information (CSI) acquisition system.

**Figure 4 sensors-19-02335-f004:**
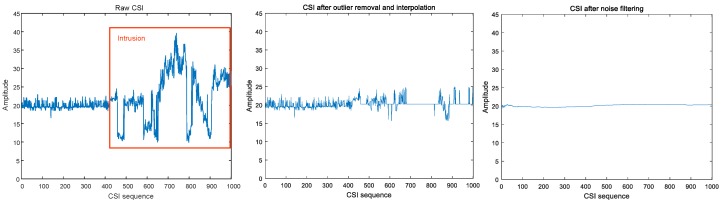
Intrusion in indoor static environment.

**Figure 5 sensors-19-02335-f005:**
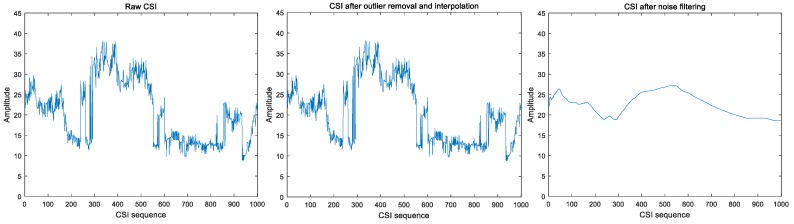
One individual walking continuously indoors.

**Figure 6 sensors-19-02335-f006:**
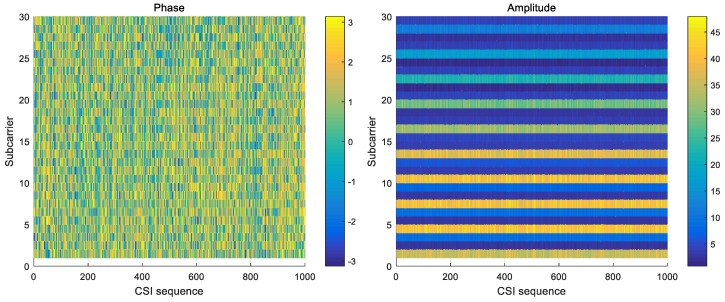
The phase and amplitude of 1000 packets in a static environment.

**Figure 7 sensors-19-02335-f007:**
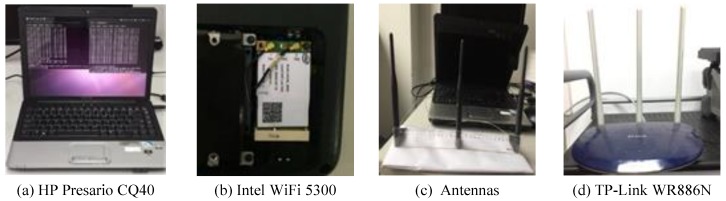
Devices.

**Figure 8 sensors-19-02335-f008:**
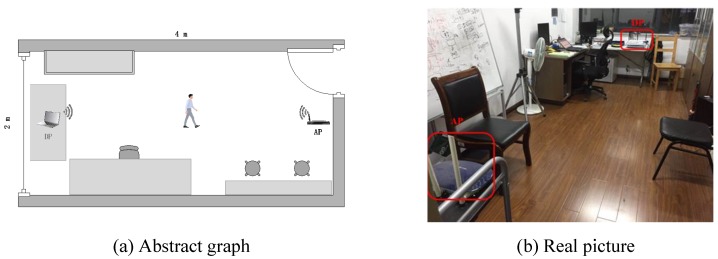
Abstract graph and real picture of experiment scenario.

**Figure 9 sensors-19-02335-f009:**
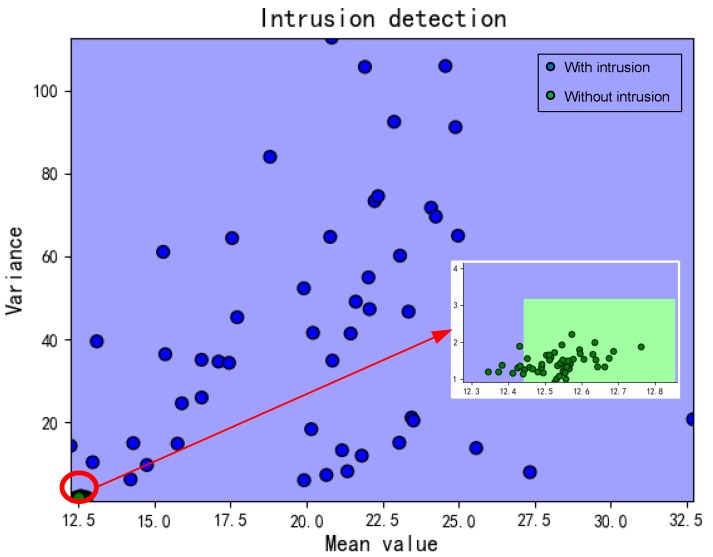
Detection results.

**Figure 10 sensors-19-02335-f010:**
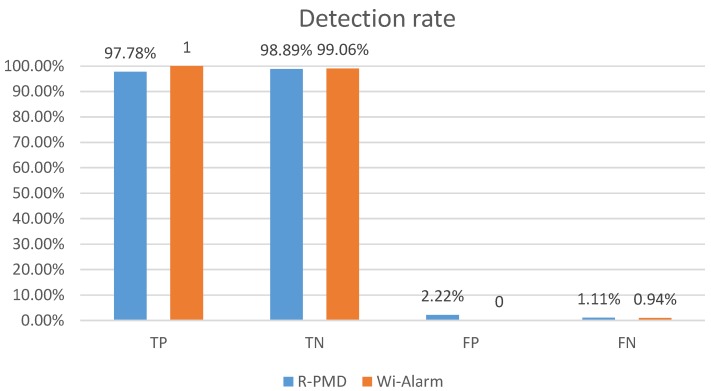
Detection rates of two systems.

**Figure 11 sensors-19-02335-f011:**
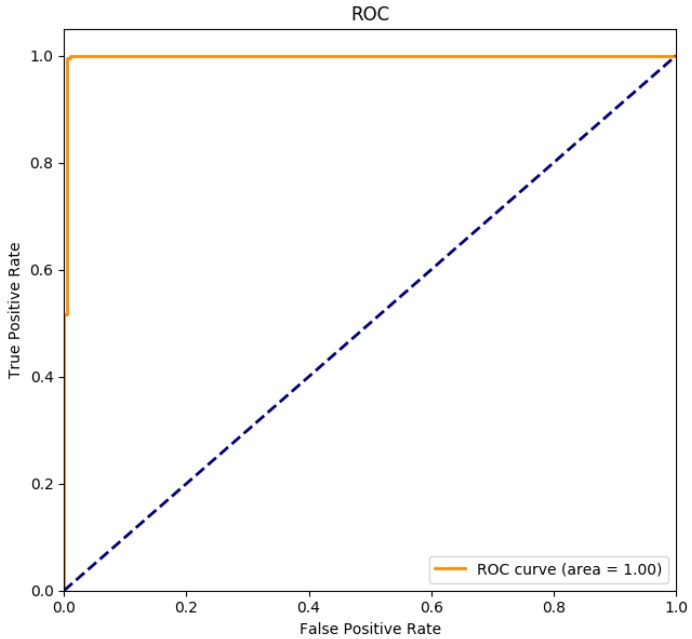
Receiver operating characteristic (ROC) curve of Wi-Alarm.

**Figure 12 sensors-19-02335-f012:**
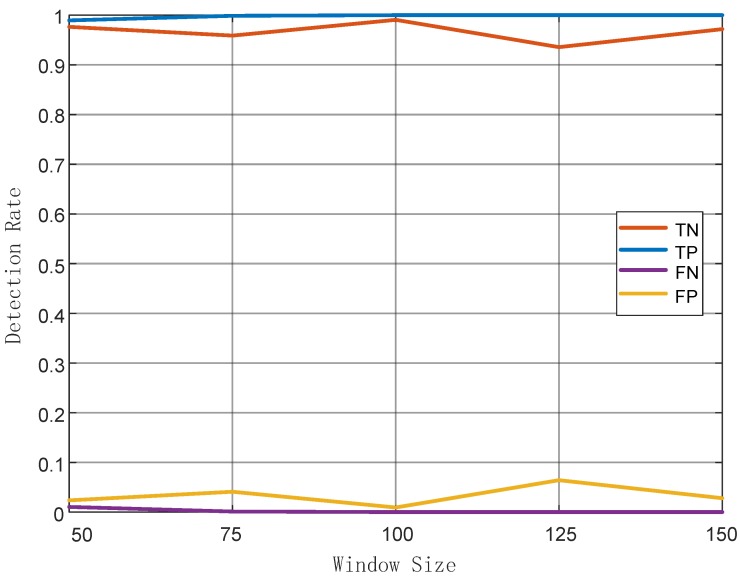
Relationship of detection rate and sliding window size.

**Figure 13 sensors-19-02335-f013:**
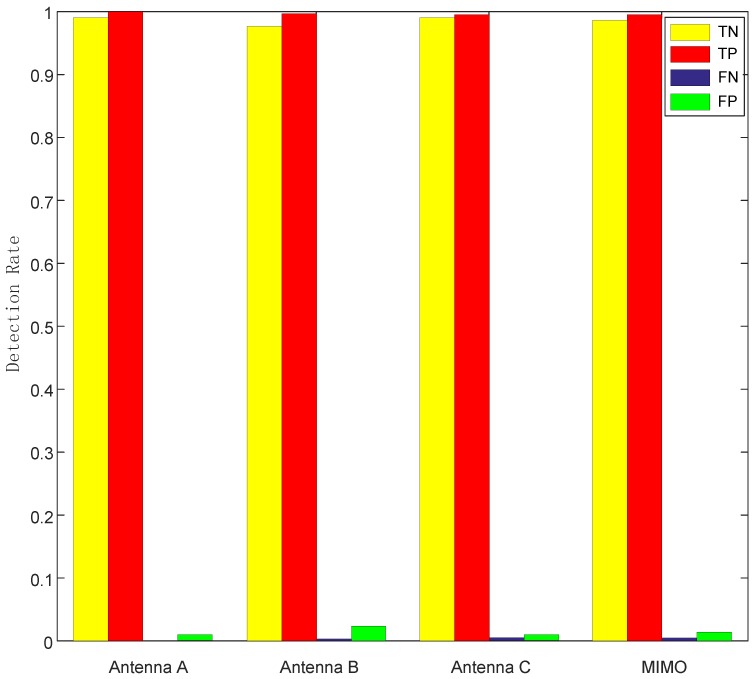
Relationship between detection rate and different antennas.

**Figure 14 sensors-19-02335-f014:**
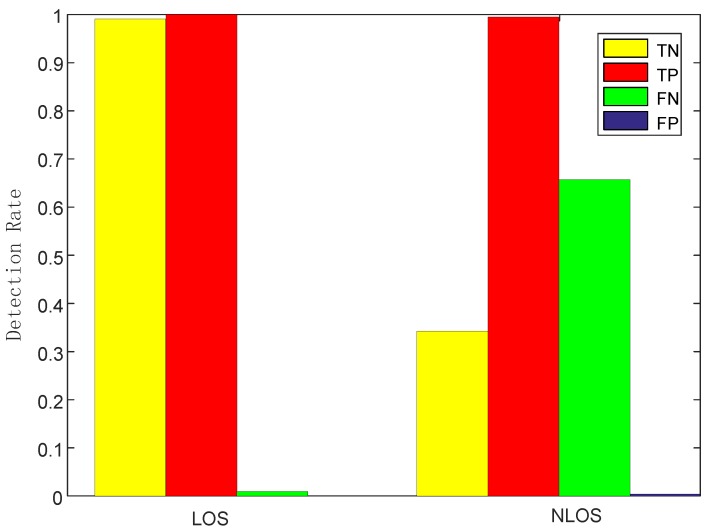
Detection rates of line-of-sight (LOS)/NLOS.

**Figure 15 sensors-19-02335-f015:**
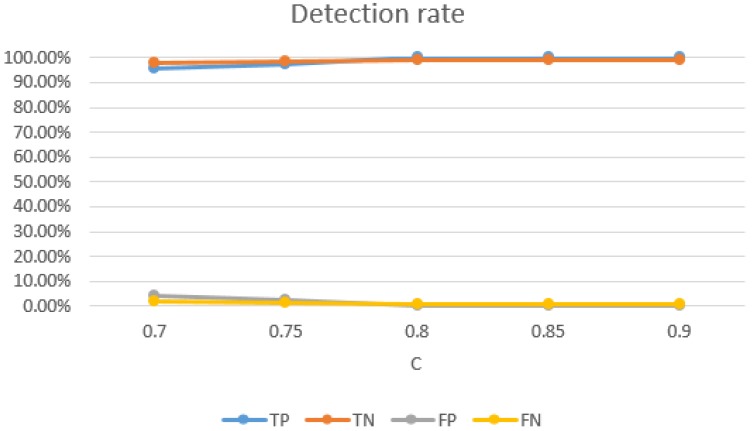
Relationship between detection rate and *C* (γ=20).

**Figure 16 sensors-19-02335-f016:**
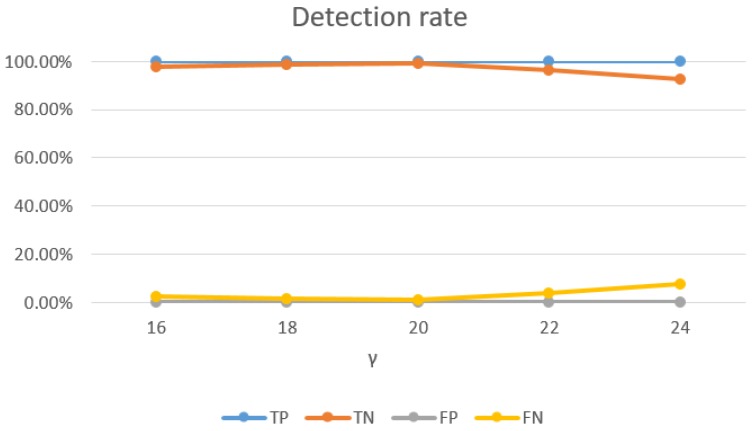
Relationship between detection rate and γ (C=0.8).

**Table 1 sensors-19-02335-t001:** Devices used in Wi-Alarm.

Label	Caption	Remarks
(a)	HP Presario CQ40	Role: DP; OS: Ubuntu 11.04; Software: CSI Tool.
(b)	Intel WiFi 5300	Equipped in Figure 7a.
(c)	Antennas	Antennas of Figure 7b; Isometry arranged.
(d)	TP-Link WR886N	Role: AP; One fixed antenna is used in data collection.

**Table 2 sensors-19-02335-t002:** Runtimes of two systems.

	Feature Calculation (Including Data Pre-Processing)	Classification
R-PMD	198.773 s	0.5 s
Wi-Alarm	0.238 s	0.06 s

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
