# Peer review of "Wi-Alarm: Low-Cost Passive Intrusion Detection Using WiFi"

_sensors, 2019, doi:10.3390/s19102335_

Reviewer 1 Report

The authors present a WiFi-based intrusion detection system called Wi-Alarm. The good points are providing some results and describing the experimental settings.

Major issues:

- The paper needs to check their writing, i.e., 

This is followed by the methodology of our system and detailed design -> the detailed design

This work needs a native English speaker to help rephrase some parts.

- 'we are the first to omit data pre-processing and extract features from raw CSI directly for intrusion detection' -> this statement could be deleted, as no information supports.

- The related work part is short, which could be improved by adding more recent studies.

e.g., A Robust Passive Intrusion Detection System with Commodity WiFi Devices. J. Sensors 2018: 8243905:1-8243905:12 (2018)

In addition, as this work uses machine learning in IDS, it could introduce some studies on machine learning based IDS.

- Section 4.4 is rather simple, it is unclear why to choose SVM for this purpose, and not other classifiers like decision tree?

- Table 2 shows runtime of two systems, it is more interesting to provide a comparison of false rates as well.

Author Response

Response to Reviewer 1 Comments

General comments

The authors present a WiFi-based intrusion detection system called Wi-Alarm. The good points are providing some results and describing the experimental settings.

We thank the reviewer for the much effort contributed for reviewing and offering feedback, which greatly motivated us to improve the quality of this paper.

We have revised the draft significantly according to the suggestions. In the revised draft, we have marked the corresponding revision in red for the sake of clarity.

Point 1: The paper needs to check their writing, i.e.,

This is followed by the methodology of our system and detailed design -> the detailed design

This work needs a native English speaker to help rephrase some parts.

Response 1:

We thank the reviewer for this comment.

The suggested revision has been implemented.

In addition, we have marked the corresponding revision in brown for the sake of clarity as follows. Please don't hesitate to let us know if there is something need to be rephrased.

1.         WiFi-based intrusion detection has superiority for its extensive coverage which means lower deployment cost.

WiFi-based intrusion detection has superiority for its extensive coverage which means low deployment cost.

2.         the raw amplitude of CSI is informative and relatively robust, by contrast, the raw phase of CSI is meaningless [12].

we think that although the raw phase of CSI is meaningless [12], the raw amplitude of CSI is informative and relatively robust.

3.         because WiFi signal works as an alarm to warn intrusion here.

because WiFi signal works as an alarm to warn people of intrusion here.

4.         CSI samples are acquired using CSI Tool [13].

CSI samples are collected with CSITool[13].

5.         We propose a simple but robust feature couple for intrusion detection where low latency is achieved.

We propose a simple but robust feature couple for intrusion detection where low latency is necessary.

6.         This is followed by the methodology of our system and detailed design in Section 4.

This is followed by the methodology of our system and the detailed design in Section 4

7.         Despite that many works have investigated CSI for device-free detection, the computational expense of the whole system is lack of study, and usually, they just discuss few influence factors. In this paper, we propose a simplest way for intrusion detection without losing accuracy. Also, different factors are considered in this paper.

Despite that many works have investigated CSI for device-free detection, the computational expense of the whole system is lack of study, and usually they just consider few influence factors. In this paper, we propose a low cost system for intrusion detection without losing accuracy. Moreover, different factors including MIMO, LOS/NLOS, size of sliding window and dynamic walking speeds are considered in this paper.

8.         The work flow of our system is as follow:

Our system works according to the following process.

9.         one person moving at three different speed

one person moving at three different speeds

10.      different cases is applied to train our SVM classifier containing static pattern and dynamic pattern.

different cases is applied to train our SVM classifier which contains a static pattern and a dynamic pattern.

Point 2: 'We are the first to omit data pre-processing and extract features from raw CSI directly for intrusion detection' -> this statement could be deleted, as no information supports

Response 2:

We thank the reviewer for this comment.

The suggested revision has been implemented as follows.

 “We focus on real-time performance of the whole system, which is neglected in previous research. We omit data pre-processing and extract features…”

Point 3: The related work part is short, which could be improved by adding more recent studies.

e.g., A Robust Passive Intrusion Detection System with Commodity WiFi Devices. J. Sensors 2018: 8243905:1-8243905:12 (2018)

In addition, as this work uses machine learning in IDS, it could introduce some studies on machine learning based IDS.

Response 3:

We thank the reviewer for this comment. We add 3 related studies in our revised manuscript and more details of each work are included as follows.

1.        “Ding et al. derived robust phase difference from the processed phase in neighbouring antennas after raw CSI preprocessing [18], extracted the covariance matrixs of normalized CSI amplitude and phase difference as features, three machine learning classification algorithms are applied for motion detection and the best results were achieved with SVM.”

Cited from [18]: Enjie Ding, Xiansheng Li, Tong Zhao, Lei Zhang, and Yanjun Hu, “A Robust Passive Intrusion Detection System with Commodity WiFi Devices,” Journal of Sensors, vol. 2018, Article ID 8243905, 12 pages, 2018. https://doi.org/10.1155/2018/8243905.

2.        “In [17], Zhou et al. applied SVC to solve the presence detection and applied SVR to solve the localization problem through regression. They applied DBSCAN on CSI data for de-noising and PCA for feature extraction. Experiments show that, compared with Bayesian algorithm, SVC performed 16.6% better in the meeting room.”

Cited from [17]: R. Zhou, X. Lu, P. Zhao, and J. Chen, "Device-Free Presence Detection and Localization With SVM and CSI Fingerprinting," IEEE Sensors Journal, vol. 17, no. 23, pp. 7990-7999, 2017.

3.        “WiGarde [16] was proposed to detect an intruder through door or window for home safety, in WiGarde, Naive Bayesian classifier was used to eliminate bad stream caused by surrounding electromagnetic noise, wavelet was used to get the width of the dynamic time window for extracting the best feature, one-class SVM was adopted to classify human intrusion.”

Cited from [16]: M. A. A. Al-qaness, F. Li, X. Ma, and G. Liu, "Device-free home intruder detection and alarm system using wi-fi channel state information," International Journal of Future Computer and Communication, vol. 5, no. 4, p. 180, 2016.

Point 4: Section 4.4 is rather simple, it is unclear why to choose SVM for this purpose, and not other classifiers like decision tree?

Response 4:

We thank the reviewer for this comment. The reasons we choose SVM are explained as follows.

“With features we extract, we need to conduct an appropriate classifier on preliminary measurements collected from several cases. There are mainly two categories of methods can be adopted for calibration free detection: threshold based and cluster based. The former distinguish different states based on the threshold value which is gained from pre-collected data. The latter classifies different clusters as different states by comparing the centre distance of each cluster. Although environment calibration and threshold training are avoided in the cluster based method, it assumes that in each group of measurements, there are at least two states involved. Otherwise, it would lead to one cluster or several clusters corresponding to a same state, which means miss or false detection. Consequently, the threshold based scheme is adopted in Wi-Alarm.

Support vector machine (SVM), as a low cost threshold based classification, is one of the most popular machine learning techniques. A considerable number of CSI based detection systems [8, 14, 16-18] have applied SVM to classify different states and achieved high accuracy. Experiments show that SVM outperforms other machine leaning method like KNN [17] and Bayesian algorithm [18]. Therefore, SVM algorithm is adopted. In this paper, we adopt the radial basis function kernel (RBF kernel, a.k.a. Gaussian kernel) as the kernel function so that a non-linear regression model can be constructed.”

Point 5: Table 2 shows runtime of two systems, it is more interesting to provide a comparison of false rates as well.

Response 5:

“The detection results of these two systems are shown in Figure 10. Both of these two systems achieve high accuracy, the TP and TN rates of Wi-Alarm are higher, which are up to 100% and 99.06% respectively.”

Reviewer 2 Report

The submission introduces a novel WiFi-based intrusion detection system aiming on discover attacks against vulnerabilities related with the Channel State Information (CSI), that for example, may lead to estimate human motions. Although the paper presents a strong research that targets a very interesting subject, this reviewer suggests the following modifications prior to be published:

·        The attack vectors to be detected by the proposal are not clearly described. A subsection within “Related Works” must explain in-depth the covered intrusions modus operandi and its characterization as monitored features.

·        How the proposal adapts to the non-stationarity/non-linearity (as concept drift) inherent in the emerging network environment.

·        Although the paper properly describes the solution architecture, the indication of the design steps adopted for its development is missing. Topics like principal/secondary objectives, null/alternative research hypothesis, assumptions, limitations, etc; should be covered by the submission.

·        Is the proposal robust against adversarial machine learning evasion tactics?

·        A sensitive-based evaluation may be more enlightening that just a TP/FP based study. Maybe to include some ROC-based analysis and/or the optimal sensor calibration criteria (e.g. optimization of the youden index) would increase the impact of the submission.

The paper presents some grammar mistakes that should be fixed.

Author Response

Response to Reviewer 2 Comments

General comments

The submission introduces a novel WiFi-based intrusion detection system aiming on discover attacks against vulnerabilities related with the Channel State Information (CSI), that for example, may lead to estimate human motions.

We thank the reviewer for the much effort contributed for reviewing and offering feedback, which greatly motivated us to improve the quality of this paper. In this paper, we take advantage of CSI to capture changes in WiFi channels. An intrusion detection system is proposed mainly to distinguish two states (human presence and absence), aiming to reduce the computational cost of the whole system without reducing detection accuracy.

Although the paper presents a strong research that targets a very interesting subject, this reviewer suggests the following modifications prior to be published.

We have revised the draft significantly according to the suggestions. In the revised draft, we have marked the corresponding revision in blue for the sake of clarity.

Point 1: The attack vectors to be detected by the proposal are not clearly described. A subsection within “Related Works” must explain in-depth the covered intrusions modus operandi and its characterization as monitored features.

Response 1:

We thank the reviewer for this comment. We revise the section “Related Work” as follows. More details of each cited research are added to clarify the covered intrusions modus operandi and its characterization as monitored features.

a)        “Kosba et al. proposed a system named RASID [5], in RASID, the sample variance of RSS is used as the selected feature, then non-parametric statistical anomaly and profile update techniques are applied to capture changes in the environment. RASID is robust, but the sampling rate of 1 sample/s is too low, resulting in too long time period for data collection.”

b)        “Yoshida et al. collected RSSI values at existing WiFi devices inserted with clock-synchronized Raspberry Pi device [14], with the outliers removed, pre-processed RSSIs are applied to two regression-based approaches for estimating the number of people in the scenario. The non-real-time experimental results showed that support vector regression-based method is better for estimating the presence/absence of people than linear regression-based method.”

c)        “Depatla et al. proposed a crowd counting system using only one pair of PC and AP with 802.11g WiFi cards, the system is based on the mathematical expression they deduced where the impact of human movement on RSSI values is divided into two parts: blocking LOS and scattering effects [15]. KL divergence between the theoretical and experimental PMF of RSSI is applied to estimate the total number of people.”

d)       “R-PMD (Robust Passive Motion Detection) [6] passes the CSI sequence through a data preprocessing module, then extracts the variance distribution of the CSI sequence and utilizes the earth mover's distance (EMD) to infer the degree of abnormality.”

e)        “SIED (Speed Independent device-free Entity Detection) [7] extracts the distribution of the variance of variances of CSI among all the subcarriers as feature and leverages a probability technique Hidden Markov Model (HMM) as the classifier to make it more accurate in human detection, it performs well when the moving speed is very slow.”

f)         “Kun et al. firstly employed a liner transformation to eliminate the significant random noise in CSI phase in PADS (PAssive Detection of moving humans with dynamic Speed) [8], it uses the maximum eigenvalues of covariance matrixes from normalized amplitude and phase as features, then SVM is introduced for different states estimation, PADS is able to accurately detect human movements of dynamic speed.”

g)        “[9] uses the same transformation as in [8] to eliminate the shift of phases of different subcarriers, it introduces the effect size to measure the change of phase as a feature and it defines two reference points for the short-term case (SES) and the long-term case (LES) to detect if someone is walking in indoor room and if someone is walking continuously respectively. However, the walking area is limited in this system, so technically, detection targets are not free.”

Point 2: How the proposal adapts to the non-stationarity/non-linearity (as concept drift) inherent in the emerging network environment.

Response 2:

We thank the reviewer for this comment. Our proposal adapt to the non-stationarity by extracting features from CSI we collected during our training phase.

Wi-Alarm contains two phases: a training phase and a monitoring phase. During the training phase, a great amount of CSI data under different cases need to be collected to enhance the robustness of the whole system, trying to cover all kinds of unexpected situations in the monitoring phase. In our system, we focus on dynamic walking speeds, which have great influence on the final detection results. Besides, we made experiments on whether the use of WiFi for Internet access (watching videos, mobile games, etc.) would affect the performance of our system. Experimental results prove that it doesn’t have much effect on system performance.

Unfortunately, our system need to be trained again when implemented in a new environment for the best performance. We will study on how to get rid of the influence of the environment and make the system self-adapted to different test environments in future.

Point 3: Although the paper properly describes the solution architecture, the indication of the design steps adopted for its development is missing. Topics like principal/secondary objectives, null/alternative research hypothesis, assumptions, limitations, etc; should be covered by the submission.

Response 3:

We thank the reviewer for this comment. Generally, we aim to propose a low-latency intrusion detection system with high accuracy. Therefore, the main indicators of the whole system are the performances of real-time and accuracy. In order to improve the real-time performance, we devote much effort to reduce the computational cost of each step without reducing detection accuracy.

We add the indication of each step of our methodology in Section 4.

In 4.1:

“In this part, the length of sliding window is significant. If the size is too small, the CSI samples are too few to calculate an effective feature, which will reduce the detection accuracy. Otherwise, larger size means more time to collect more CSI samples, undermining the real-time performance of the system. In our system, through experiment analysis, it is appropriate to set to transmit 100 packaged per second, i.e., CSI sampling frequency is 100Hz.”

In 4.2:

“In Wi-Alarm, we extract features from raw CSI data without preprocessing in order to save computational cost for better real-time performance. There is no doubt that omitting preprocessing will reduce much computational cost, in this part, we will discuss the feasibility of preprocessing omission.”

In 4.3:

“Feature extraction plays an important role in intrusion detection. The features we choose should meet the following requirements:

The features should be sensitive to human presence. The features extracted from the CSI of static environment should be quite different from the ones extracted from the CSI with human motion.

The features should be simple to calculate. For real-time consideration, less computation cost on feature extraction would be better.”

In 4.4:

“Support vector machine (SVM), as a low cost threshold based classification, is one of the most popular machine learning techniques. A considerable number of CSI based detection systems [8, 14, 16-18] has applied SVM to classify different states and achieved high accuracy. Experiments show that SVM outperforms other machine leaning method like KNN [17] and Bayesian algorithm [18]. Therefore, SVM algorithm is adopted for satisfying the system requirements for real-time and high detection accuracy.”

Point 4:  Is the proposal robust against adversarial machine learning evasion tactics?

Response 4:

We thank the reviewer for this comment. We have investigated a number of literatures on WiFi-based detection including intrusion detection, crowd counting and activity recognition. The research on this aspect is still blank. This would be a future direction of research and challenge to be faced. We will improve this part in our future work.

Point 5: A sensitive-based evaluation may be more enlightening that just a TP/FP based study. Maybe to include some ROC-based analysis and/or the optimal sensor calibration criteria (e.g. optimization of the youden index) would increase the impact of the submission.

Response 5:

We thank the reviewer for this comment. We add ROC-based analysis in Section 5.2.

R-PMD [6] is repeated in our experiments. The detection results of these two systems are shown in Figure 10. Both of these two systems achieve high accuracy, the TP and TN rates of Wi-Alarm are higher, which are up to 100% and 99.06% respectively.

 In addition, the receiver operating characteristic (ROC) curve of our classifier is shown in Figure 11, the result proves that the classifier we choose is trained well and achieves excellent performance.

Figure 11. ROC curve of Wi-Alarm.

Actually, during our explorations, we also tried using Doppler shift as a feature to train SVM classifier, the ROC curve of this classifier is shown below, which is much worse than the SVM in Wi-Alarm. This also illustrates the importance of feature selection.

Using Doppler Shift as a feature

Point 6: The paper presents some grammar mistakes that should be fixed.

Response 6:

We thank the reviewer for this comment. We have marked the corresponding revision in brown for the sake of clarity. Please don't hesitate to let us know if there is something need to be rephrased.

1.         WiFi-based intrusion detection has superiority for its extensive coverage which means lower deployment cost.

WiFi-based intrusion detection has superiority for its extensive coverage which means low deployment cost.

2.         the raw amplitude of CSI is informative and relatively robust, by contrast, the raw phase of CSI is meaningless [12].

we think that although the raw phase of CSI is meaningless [12], the raw amplitude of CSI is informative and relatively robust.

3.         because WiFi signal works as an alarm to warn intrusion here.

because WiFi signal works as an alarm to warn people of intrusion here.

4.         CSI samples are acquired using CSI Tool [13].

CSI samples are collected with CSITool[13].

5.         We propose a simple but robust feature couple for intrusion detection where low latency is achieved.

We propose a simple but robust feature couple for intrusion detection where low latency is necessary.

6.         This is followed by the methodology of our system and detailed design in Section 4.

This is followed by the methodology of our system and the detailed design in Section 4

7.         Despite that many works have investigated CSI for device-free detection, the computational expense of the whole system is lack of study, and usually, they just discuss few influence factors. In this paper, we propose a simplest way for intrusion detection without losing accuracy. Also, different factors are considered in this paper.

Despite that many works have investigated CSI for device-free detection, the computational expense of the whole system is lack of study, and usually they just consider few influence factors. In this paper, we propose a low cost system for intrusion detection without losing accuracy. Moreover, different factors including MIMO, LOS/NLOS, size of sliding window and dynamic walking speeds are considered in this paper.

8.         The work flow of our system is as follow:

Our system works according to the following process.

9.         one person moving at three different speed

one person moving at three different speeds

10.      different cases is applied to train our SVM classifier containing static pattern and dynamic pattern.

different cases is applied to train our SVM classifier which contains a static pattern and a dynamic pattern.

Round  2

Reviewer 1 Report

There are still some major issues:

- SVM parameters were not discussed in the evaluation, e.g., C = 0.8 and g = 20.

- It is unclear how the authors performed the attacks, and it is important to explore scalability.

- Whether this scheme can detect some types of attacks? Limitations should be discussed in a proper way in a subsection.

Author Response

Response to Reviewer 1 Comments 

(Round 2)

General comments

       There are still some major issues:

We thank the reviewer for the much effort contributed for reviewing and offering feedback, which greatly motivated us to improve the quality of this paper.

We have revised the draft significantly according to the suggestions. In the revised draft, we have marked the corresponding revision in red for the sake of clarity.

Point 1: SVM parameters were not discussed in the evaluation, e.g., C = 0.8 and g = 20.

Response 1:

We thank the reviewer for this comment.

We add the evaluation of SVM parameters in Subsection 5.2.

Impact of SVM parameters: The selection of SVM parameters also has significant impact on detection rate of Wi-Alarm. Through our experiment evaluations, on one hand, higher detection rates are achieved with bigger C, as Figure 15 shown. However, C controls overfitting of the model, so it should not be too big and 0.8 is a good setting. On the other hand, as Figure 16 shown, the TN rate decreases with the increase of γ. Meanwhile, γcontrols the degree of nonlinearity of the model and biggerγis preferred in this respect. Therefore, 20 is a good setting forγ. In conclusion, we set C=0.8 andγ=20 for the best performance of Wi-Alarm.”

Point 2: It is unclear how the authors performed the attacks, and it is important to explore scalability.

Response 2:

We thank the reviewer for this comment.

As mentioned in Section 1: “Intrusion detection is a dynamic process of monitoring whether there exists any entity breaking into a given area, then making alarms if necessary.”

We add related information in “Conclusion”:

“In our system, attacks are performed when an intruder is entering the monitoring area unexpectedly wherever he comes from. Wi-Alarm is proposed mainly to distinguish two states (human presence and absence) and detect whether there exists any people in the area of interest. It could be applied in the field of safety monitoring.”

Point 3: Whether this scheme can detect some types of attacks? Limitations should be discussed in a proper way in a subsection.

Response 3:

We thank the reviewer for this comment.

Our system can detect the type of attack of one intruder is entering the monitoring area unexpectedly when the environment is static without any human motion (no one exists). During the training phase, a great amount of CSI data under different cases need to be collected to enhance the robustness of the whole system, trying to cover all kinds of unexpected situations in the monitoring phase. In our system, we focus on dynamic walking speeds, which have great influence on the final detection results. So our system is robust to dynamic walking speeds of intruders no matter how slowly he moves.

 The limitations are added in the Section “Conclusion”:

 “However, there still exists some limitations. Firstly, when installed in a different scenario, our system need to be trained again to find a suitable threshold. Moreover, usually there already exists someone in the given area which means the original environment isn't static before any intrusion occurs. Consequently, we should explore new features to model the motion introduced by the intruder coming from the outside aiming to distinguish it from the motion inside.

Future work will focus on exploring novel and effective features to enhance the adaptability of our system and the functions of detecting more types of attacks.”
